# A trial-based cost-utility analysis of sugemalimab vs. placebo as consolidation therapy for unresectable stage III NSCLC in China

Wei Li[ID]*, Li Wan

Department of Pharmacy, Maternal and Child Health Hospital of Hubei Province, Tongji Medical College, Huazhong University of Science and Technology, Wuhan, China

* liwei1988hust@126.com

## Abstract

### Objective

The effectiveness of sugemalimab vs. placebo in post-chemoradiotherapy patients with locally advanced, unresectable stage III NSCLC has been demonstrated and approved by China National Medical Products Administration. The purpose of this study was to assess the cost-effectiveness of sugemalimab vs. placebo for consolidation treatment of stage III NSCLC from the perspective of the Chinese healthcare system.

### Methods

A 3-state Markov model with a 3-week cycle length was performed to appraise the incremental cost-utility ratio (ICUR) of sugemalimab consolidation therapy based on the GEM-STONE-301 clinical trial over a 10-year time horizon. Only direct medical costs, including costs of drug (maintenance and subsequent treatment), routine follow-up, best supportive care, and terminal care in end of life were considered in this model. Costs and health utilities were obtained from local databases and published articles. Sensitivity and scenario analyses were adopted to evaluate the model uncertainty. Internal and external data sources were used to justify the plausibility of the extrapolated portion of the survival model chosen.

### Results

In comparison with the placebo, sugemalimab consolidation therapy was not cost-effective as it yielded an ICUR value of $90,277 and $49,692 for the concurrent chemoradiotherapy (cCRT) and the sequential chemoradiotherapy (sCRT) population at the willingness-to-pay (WTP) threshold of $37,663/QALYs, respectively. When taking the sugemalimab patient assistance program (PAP) into consideration, sugemalimab consolidation therapy was cost-effective with an ICUR dramatic decreases below the WTP. Sensitivity analyses demonstrated that the ICUR was most sensitive to the discount rate and subsequent treatment. However, none of the sensitive parameters could affect the cost-effective conclusions without or with PAP. Scenario analyses revealed that the model was particularly affected by

**Data Availability Statement:** All relevant data are within the paper and its Supporting information files.

**Funding:** This work was supported by the Maternal and Child Health Hospital of Hubei Province Research Project (No. 2021SFYM030). The funders had no role in study design, data collection and analysis, decision to publish, or preparation of the manuscript.

**Competing interests:** The authors have declared that no competing interests exist.

assumptions regarding discount in sugemalimab, time horizon, mean duration of sugemalimab maintenance treatment.

## Conclusions

From the perspective of Chinese healthcare system, sugemalimab consolidation therapy was not a cost-effective strategy in cCRT and sCRT patients with unresectable stage III NSCLC. Given that the sugemalimab PAP was available, sugemalimab consolidation therapy became a cost-effective option.

## 1. Introduction

Lung cancer is the most common malignancy and the leading cause of tumor-related death in China [1]. Non-small cell lung cancer (NSCLC) is the most prevalent subtype of lung cancer, and approximately 30% of newly diagnosed patients are at an advanced unresectable stage (stage III) [2]. For a decade, platinum-based chemotherapy, concurrent with radiotherapy (cCRT), was the standard of care for patients with stage III NSCLC all over the world until the approval of durvalumab, which obtained satisfactory clinical benefit in the PACIFIC trial and was approved for whose disease had not progressed after cCRT [3,4]. However, the PACIFIC trial restricted enrollment to patients after cCRT rather than sequential chemoradiotherapy (sCRT). It should be concerned that a number of clinical trials and meta-analysis have shown that treatment-related toxicities are significantly increased with cCRT compared to sCRT [4–6]. Furthermore, comorbidity, a key predictor of poor prognosis in different cancers [7], may also deteriorate patients' performance status often more than the tumor development [8]. Recently, a study indicated that 32.2% had at least one comorbid condition among lung cancer patients in China [9]. As a matter of fact, due to tolerability and Eastern Cooperative Oncology Group (ECOG) performance status issues, many patients therefore chose sequential chemoradiotherapy (sCRT) as an option strategy [9,10]. As an alternative to cCRT, sCRT is recommended for patients who cannot tolerate or access cCRT in international treatment guidelines [11–13] and widely used in clinical practice throughout the world. Currently in the United Kingdom (UK), 55% of stage III NSCLC patients are treated with sCRT, compared with 45% who receive cCRT [14]. Although sCRT is widely used in clinical practice, whether the use of immune checkpoint inhibitors after sCRT prolongs survival in patients with unresectable stage III NSCLC is unknown.

Sugemalimab (suge), a full length, fully human, anti-programmed death ligand 1 (PD-L1) monoclonal antibody, is being developed by CStone Pharmaceuticals. In the phase III GEM-STONE-301 trial (NCT03728556), sugemalimab vs. placebo significantly prolonged the median progression-free survival (PFS) and median overall survival (OS) [15]. At data cutoff (March 1, 2022) for the final analysis [16,17], the median PFS was 15.7 vs. 8.3 months (HR = 0.71) and 8.1 vs. 4.1 months (HR = 0.57) in cCRT and sCRT population, respectively. Furthermore, the median OS was not reached (NR) vs. 32.4 months (HR = 0.75) and NR vs. 24.1 months (HR = 0.60) in cCRT and sCRT population, respectively. Based on its significant clinical benefits and acceptable safety profile, sugemalimab was approved in China for consolidation treatment in patients with stage III NSCLC [18].

Despite its clinical benefits, the huge cost of sugemalimab consolidation therapy might bring potential economic impact to patients and their families, mirroring the efficiency issue in a health recourse-limited setting. Previous studies have assessed the cost-effectiveness of sugemalimab in first-line treatment of metastatic NSCLC (stage IV) [19,20]. However, the

economic evaluation of sugemalimab therapy in different clinical setting (i.e. stage III NSCLC) remains to be performed. This study aimed to provide a comprehensive economic evaluation of sugemalimab consolidation therapy for unresectable stage III NSCLC in the context of the Chinese health care system.

## 2. Methods

### 2.1 Model structure and outcomes

A mathematical model that combined decision tree and Markov model was constructed to evaluate the clinical and economic outcomes of sugemalimab vs. placebo, as a consolidation therapy in patients with locally advanced, unresectable, stage III NSCLC (Fig 1). A 3-state Markov model, including three mutually exclusive disease-related health states: progression-free survival (PFS), progressed disease (PD), and death was performed by using TreeAge Pro 2020 (TreeAge Software, Williamstown, MA) (Fig 1). It was assumed that all patients entered the model in the PFS state and then either remained in the same state or transited to the other states during a Markov cycle length of 3 weeks, which was in line with the treatment cycle length. The initial age of the simulated cohort population was set to 60 years old, which was in line with the average age of the GEMSTONE-301 trial. The time horizon for the model was 10 years.

The primary model outcomes were the corresponding total costs of two therapeutic regimens, quality-adjusted life years (QALYs), and the incremental cost utility ratio (ICUR). Both costs and utility values were discounted at an annual rate of 5% for base-case analysis, according to guideline for health economic evaluations in China [21]. All costs were converted into 2021 US dollars (US 1$ = CNY ¥6.45). Three times of the per capita gross domestic product (GDP) of China in 2021 (US$37,663/QALYs) was used as the willingness-to-pay (WTP) threshold to evaluate the cost-effectiveness of the two competing strategies [21].

### 2.2 Clinical data and transition probabilities

Clinical efficacy and safety data were obtained from GEMSTONE-301 trial [15] which recruited 381 Chinese patients from 50 hospitals or academic research centres between Aug 30, 2018 and Dec 30, 2020, were randomly assigned to sugemalimab (n = 255) and to placebo

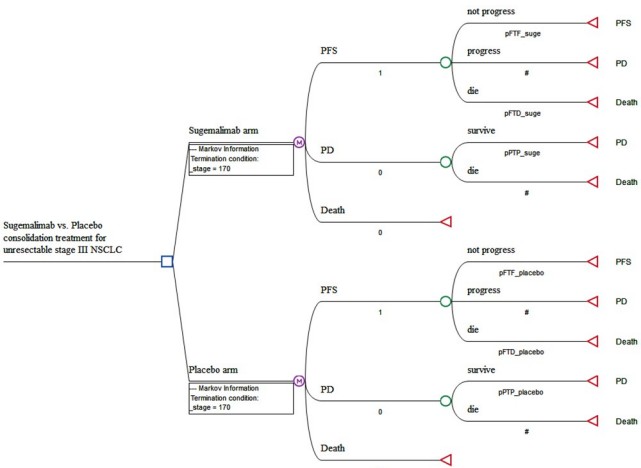

**Fig 1. The schematics of the decision tree and the Markov state transition model.**

(n = 126). Patients received a continuous and fixed dose of sugemalimab (1200mg, every 3 weeks) or placebo as consolidation therapy for up to 24 months, unless confirmed disease progression, unacceptable toxicity or withdrawal of consent. The time-dependency transition probabilities of patients remaining in PFS and PD state were simulated based on the PFS and OS Kaplan-Meier (K-M) curves of the GEMSTONE-301 trial, data cutoff at March 1, 2022, respectively [16,17]. First, GetData Graph Digitizer software (version 2.26) was employed to extracted the graphical data from published K-M curves. Second, R software (version 4.0.3) was used to reconstruct the individual patient data (IPD) derived by Hoyle et al [22]. Third, to extrapolate survival curve beyond the observation period, we considered the following parametric survival functions to fit the IPD: Exponential, Weibull, Gompertz, Log-logistic, Log-normal, and parametric mixture cure models. The choice of the model to use was based on Akaike's information criterion (AIC), Bayesian information criterion (BIC), visual inspection and clinical plausibility of the extrapolations. Finally, the best fitting distribution parameters for both two arms were shown in Table 1 and the goodness-of-fit results were shown in S1 Table. We also provided the exploration and fitting of PFS and OS curves in comparison with external data for visual inspection (see Figs 2 and 3). To calculate the probability of transition from the PFS state to death state, we assumed that it was equal to the age-specific background mortality derived from the National Bureau of Statistics of China (2019) [23]. The method for calculating transition probabilities was detailed in the published article [24].

## 2.3 Cost and utility

Only direct medical costs, calculated from the perspective of the Chinese healthcare system, including costs of drugs (maintenance and subsequent treatment), best support care (BSC), routine follow-up, and terminal care in end of life. Due to the low incidence of grade 3 to 4 serious adverse events (SAEs) in both arms (less than 2%), we did not consider the costs and disutility values related to SAEs. In calculating dosage amounts, we assumed a 60-year-old patient with an average weight of 65 Kg, height of 1.64 m, and 70 mL/min creatinine clearance [25], resulting in a body surface area (BSA) of 1.72 $m^2$ [26]. Currently, sugemalimab patient assistance program (PAP) was conducted for eligible patients to improve the drug affordability in China [27]. The PAP supports patients to pay for 2 cycles of sugemalimab, followed by 2 cycles of free sugemalimab (2+2); and then pay for 2 cycles, followed by 25 cycles of donations (2+25); and then pay for 1 cycle, followed by 3 cycles of donations (1+3).

**Table 1. Best fit and the values of the parameters.**

| | best fitting | Scale/Rate/ Sigma | Shape/Q | Theta | Mu |
|---|---|---|---|---|---|
| **cCRT population** | | | | | |
| Suge, OS | MCM-Gamma | 0.05040 | 1.93840 | 0.35830 | - |
| Suge, PFS | Gompertz | 0.04322 | -0.03176 | - | - |
| Placebo, OS | Log-normal | 3.78600 | 1.09100 | - | - |
| Placebo, PFS | Gompertz | 0.05935 | -0.03300 | - | - |
| **sCRT population** | | | | | |
| Suge, OS | MCM-Gamma | 0.01330 | 1.17000 | 0.00217 | - |
| Suge, PFS | Gompertz | 0.07675 | -0.04160 | - | - |
| Placebo, OS | Log-logistic | 0.02662 | 1.99700 | - | - |
| Placebo, PFS | MCM-Generalised gamma | 0.41818 | -2.05638 | 0.04858 | 1.15493 |

PFS, progression-free survival; OS, overall survival; cCRT, concurrent chemoradiotherapy; sCRT, sequential chemoradiotherapy; Suge, sugemalimab; MCM, mixture cure model.

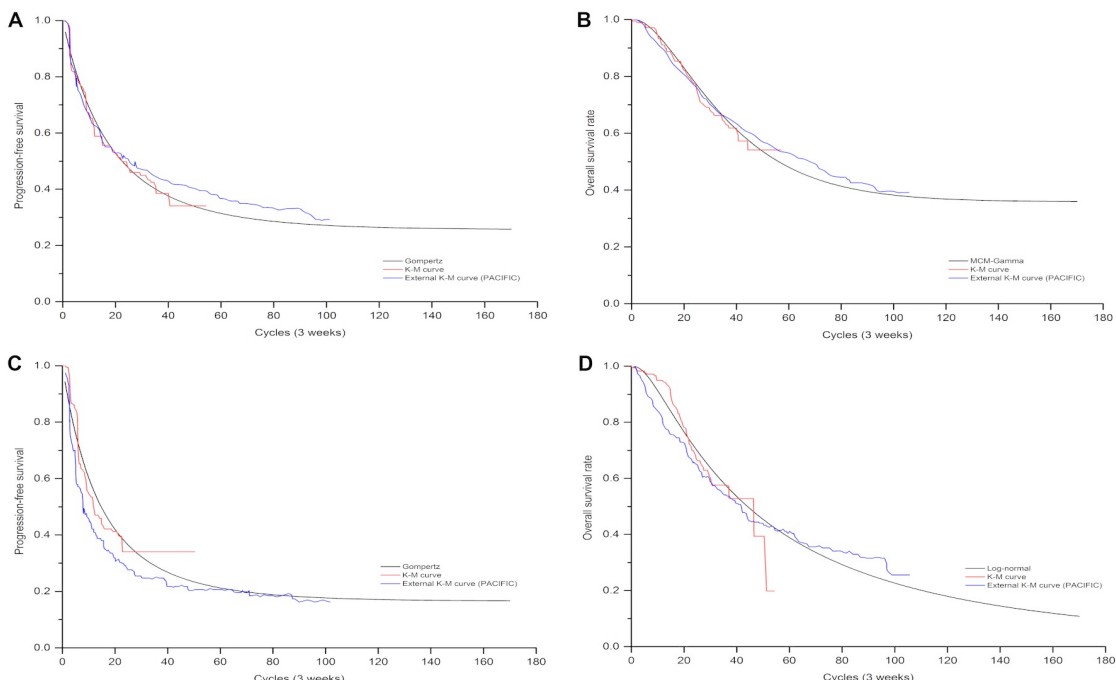

**Fig 2. The exploration and fitting of PFS and OS curves for cCRT population in comparison with external data.** Model simulation visual PFS curve (A), OS curve (B) of sugemalimab arm and PFS curve (C), OS curve (D) of placebo arm.

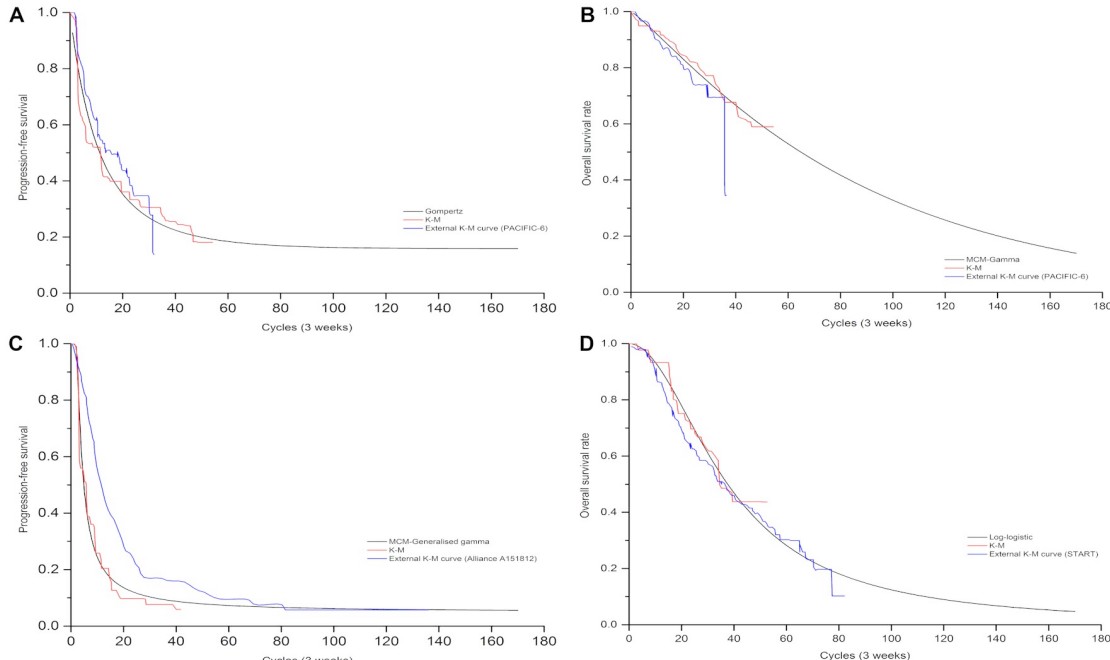

**Fig 3. The exploration and fitting of PFS and OS curves for sCRT population in comparison with external data.** Model simulation visual PFS curve (A), OS curve (B) of sugemalimab arm and PFS curve (C), OS curve (D) of placebo arm.

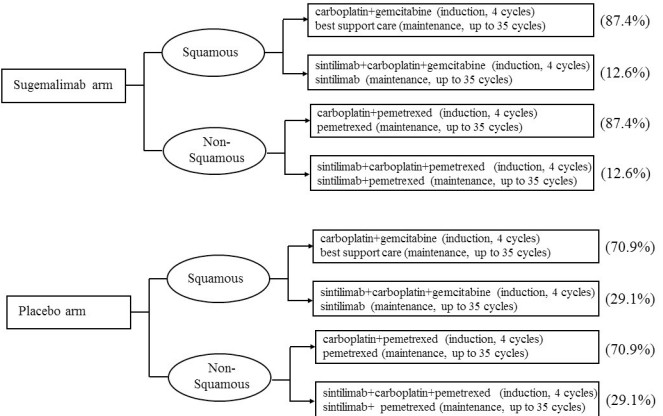

**Fig 4. The proportion of subsequent immunotherapy and detailed treatment strategies varied with tumor histology type at progression for patients with stage III NSCLC.**

As no detailed subsequent anticancer treatment data was available for GEMSTONE-301 trial at cutoff date March 1, 2022, we assumed that patients at progression would receive platinum-based chemotherapy, with or without immunotherapy according to guidelines for the clinical diagnosis and treatment of lung cancer in China [28]. The proportion of subsequent immunotherapy for sugemalimab arm and placebo arm was 12.6% and 29.1%, which was gathered from the five-year survival outcomes of PACIFIC trial [29]. Sintilimab and pembrolizumab were the preferred option for subsequent immunotherapy [15], based on the interim analysis (cutoff date: March 8, 2021) of GEMSTONE-301 trial. Because of the large price difference between the two drugs, we used sintilimab for the baseline analysis and pembrolizumab for the scenario analysis. For simplification, we assumed that patients with non-squamous (NSQ) NSCLC after progression to receive sintilimab (200mg, every 3 weeks) or placebo, plus carboplatin (AUC 5 mg/mL/min, every 3 weeks) and pemetrexed (500 mg/m$^2$, every 3 weeks), for up to 4 cycles, followed by sintilimab or placebo for up to a total of 35 cycles plus pemetrexed maintenance therapy [30]. Patients with squamous (SQ) NSCLC after progression to receive sintilimab (200mg, every 3 weeks) or placebo, plus carboplatin (AUC 5 mg/mL/min, every 3 weeks) and gemcitabine (1.0 g/m$^2$, day 1 and day 8, every 3 weeks), for up to 4 cycles, followed by sintilimab or placebo for up to a total of 35 cycles maintenance therapy [31]. The detailed subsequent treatment strategies were shown in Fig 4. In addition, a 2-year maximum treatment duration of sugemalimab, sintilimab and pembrolizumab was taken into consideration based on previous studies [32]. All costs were derived from local charge, median price of the winning bid product derived from the Chinese Drug Bidding Database by the YaoZH [33] or previously published literature.

Health utility values for each health state in our model were derived from previously published studies, as no health-related quality of life (HRQoL) was published in clinical trial. We set the health utility values estimates for PFS and PD was 0.901 and 0.863, respectively [34]. Furthermore, a half-cycle correction was implemented to the outcomes, according to the China Guidelines for Pharmacoeconomic Evaluation [21].

## 2.4 Sensitivity analysis

One-way deterministic (DSA) and probabilistic sensitivity analysis (PSA) were performed to assess the robustness of model outcomes. In one-way sensitivity analysis, relevant parameters

that had substantial impact on ICUR, were tested orderly at the upper and lower limits of plausible ranges, which were listed and illustrated in Table 2. The results of one-way sensitivity analysis were graphed in the tornado diagram. PSA was performed to determine the effects of uncertainty in all model parameters simultaneously via 1000 Monte Carlo simulations, which were illustrated in cost-effectiveness acceptability curve. The key parameters input our model were shown in Tables 1 and 2.

## 2.5 Scenario analysis

Scenario analyses were performed using variations in key model setting and assumptions, including time horizon, tumor histological type distribution, discount in sugemalimab, choice of subsequent immunotherapy, mean duration of maintenance treatment and subsequent immunotherapy.

# 3. Results

## 3.1 Base-case analysis

Discounted results for sugemalimab following cCRT or sCRT vs. placebo were presented in Table 3. Compared with the placebo arm, suge arm resulted in ICUR values of $90,277 and $49,692 for the cCRT and sCRT population, respectively (Table 3). The suge arm was clearly

**Table 2. Model inputs: Base case values, ranges, and distributions for sensitivity analysis.**

| Parameters | Base case | Range | Distribution | Source |
|---|---|---|---|---|
| **Costs (US $)** | | | | |
| Sugemalimab (600mg)[a] | 1,918.6 | 1,822.7–1918.6 | Fixed in PSA | YaoZH |
| Carboplatin (100mg) | 8.0 | 6.4–9.6 | Gamma | Local charge |
| Pemetrexed (500mg) | 424.2 | 339.4–509.0 | Gamma | YaoZH |
| Pembrolizumab (100 mg) | 2778 | 2222.4–3333.6 | Gamma | YaoZH |
| Sintilimab (100mg) | 440.8 | 352.6–529.0 | Gamma | YaoZH |
| Gemcitabine (200mg) | 6.6 | 5.3–7.9 | Gamma | Local charge |
| Routine follow-up cost per cycle[b] | 55.6 | 27.8–83.4 | Gamma | [26] |
| Terminal care per cycle[c] | 2,627.8 | 1,313.9–3,941.7 | Gamma | [26] |
| Best supportive care per cycle[d] | 337.5 | 168.75–506.25 | Gamma | [26] |
| **Utility values** | | | | |
| Utility of PFS | 0.901 | 0.883–0.918 | Beta | [34] |
| Utility of PD | 0.863 | 0.845–0.880 | Beta | [34] |
| Discount rate(%) | 5 | 0–8 | Fixed in PSA | [21] |
| **others** | | | | |
| Proportion of subsequent immunotherapy for suge arm | 0.126 | 0–0.126 | Beta | [29] |
| Proportion of subsequent immunotherapy for placebo arm | 0.291 | 0.291–1 | Beta | [29] |
| Weight (kg) | 65 | 52–78 | Normal | [26] |
| Body surface area (m$^2$) | 1.72 | 1.38–2.06 | Normal | [26] |
| Proportion of squamous cell carcinoma | 0.7 | Fixed | Fixed | [15] |

PFS, progression-free survival; PD, progressed disease; PSA, probabilistic sensitivity analysis; suge, sugemalimab.

[a]The lower limit is 95% of the price.

[b]The cost of routine follow-up included the cost of outpatient physician visit, hospitalization, and laboratory tests.

[c]The terminal care cost referred to the cost of palliative end-of-life.

[d]BSC referred to the intervention of clinical symptoms caused by cancer, including anti-inflammatory treatment, analgesic treatment, antiemetic treatment, thoracic and abdominal puncture decompression, blood transfusion and nutritional support.

**Table 3. Base-case analysis results without or with PAP.**

| Population | Treatment | Cost ($) | Incremental cost | QALYs | Incremental QALYs | ICUR |
|---|---|---|---|---|---|---|
| cCRT | | | | | | |
| without PAP | suge | 84,179 | 74,930 | 3.72 | 0.83 | 90,277 |
| | placebo | 9,249 | NA | 2.89 | NA | NA |
| with PAP | suge | 22,002 | 12,753 | 3.72 | 0.83 | 15,365 |
| | placebo | 9,249 | NA | 2.89 | NA | NA |
| sCRT | | | | | | |
| without PAP | suge | 67,987 | 54,164 | 3.43 | 1.09 | 49,692 |
| | placebo | 13,823 | NA | 2.34 | NA | NA |
| with PAP | suge | 22,448 | 8,625 | 3.43 | 1.09 | 7,913 |
| | placebo | 13,823 | NA | 2.34 | NA | NA |

cCRT, concurrent chemoradiotherapy; sCRT, sequential chemoradiotherapy; QALYs, quality-adjusted life years; ICUR, incremental cost-utility ratio; NA, not applicable; suge, sugemalimab; PAP: Patient assistance program.

not a cost-effective strategy for the cCRT and sCRT population under the Chinese cost-effective WTP threshold of $37,663/QALYs, to receive maintenance therapy with sugemalimab, in comparison of placebo. However, when the PAP was considered, the suge arm became a cost-effective strategy for both cCRT and sCRT populations (see Table 3).

## 3.2 DSA

Tornado diagrams of the top 10 most influential key variables in cCRT and sCRT population were illustrated in Figs 5 and 6, respectively. According to the one-way DSA results, the discount rate, proportion of patients received subsequent immunotherapy in placebo arm after

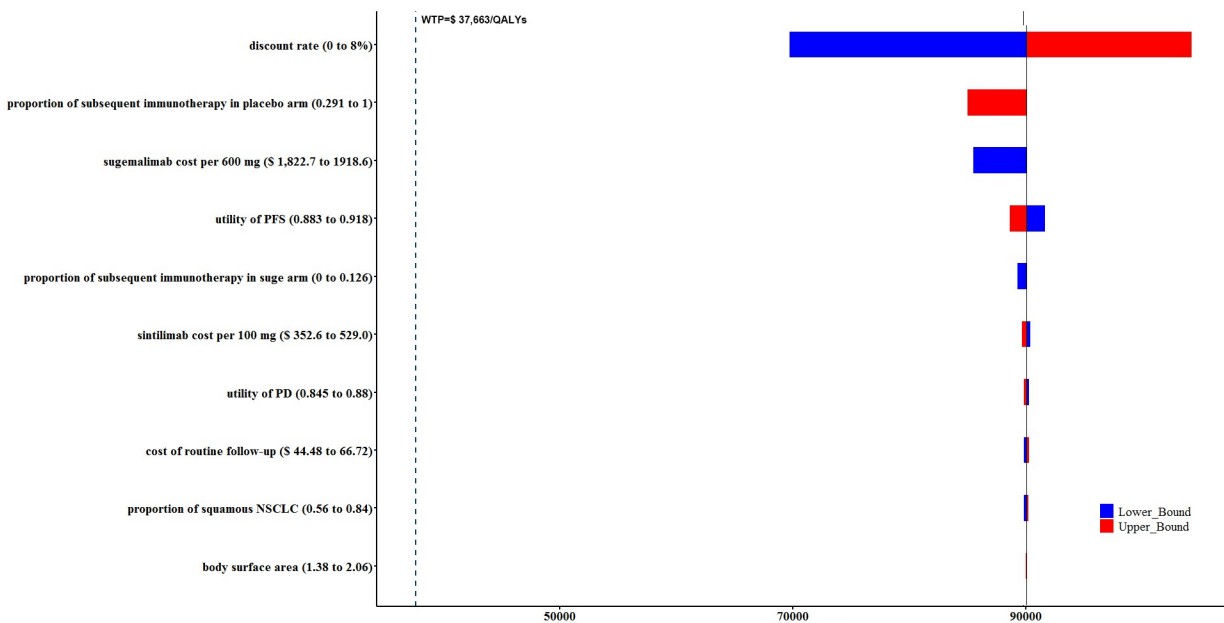

**Fig 5. Tornado diagram of one-way deterministic sensitivity analysis (DSA) of sugemalimab vs. placebo.** DSA for cCRT population without PAP. cCRT, concurrent chemoradiotherapy; PAP: Patient assistance program.

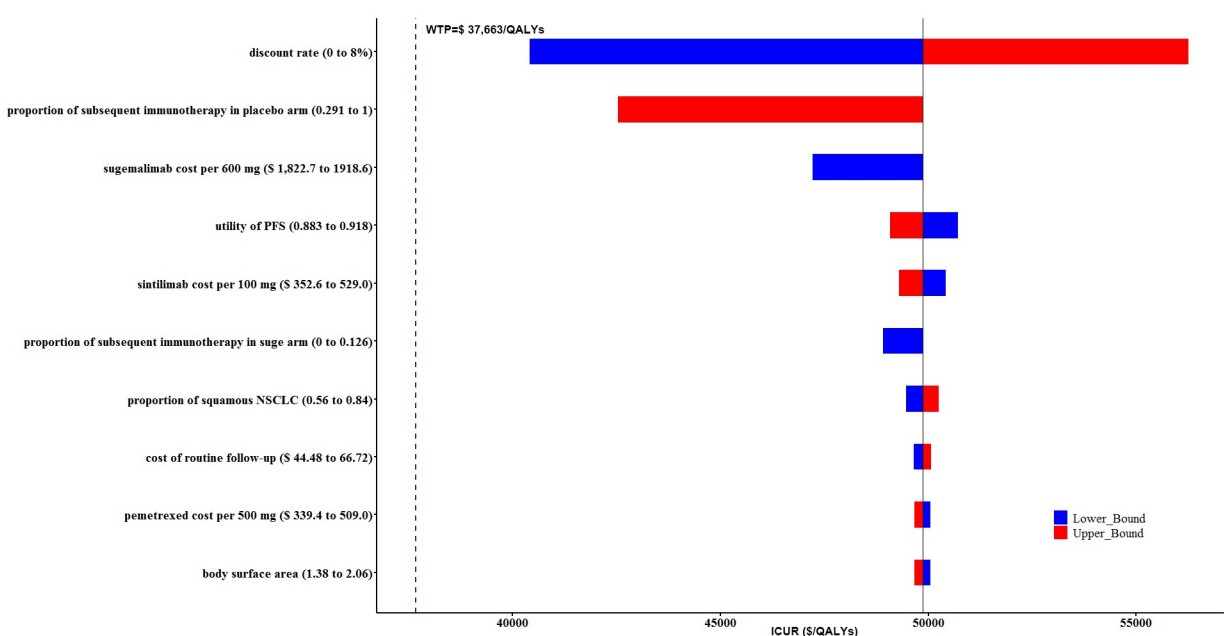

**Fig 6. Tornado diagram of one-way deterministic sensitivity analysis (DSA) of sugemalimab vs. placebo.** DSA for sCRT population without PAP. sCRT, sequential chemoradiotherapy; PAP: Patient assistance program.

disease progression, and the cost of sugemalimab were the main driving variables. However, none of the fluctuations of sensitive parameters could affect the cost-utility conclusions without (Figs 5 and 6) or with PAP (S1 and S2 Figs) for cCRT and sCRT population. On the whole, our base-case analysis results were relatively robust.

### 3.3 PSA

The results of PSA were presented in Fig 7. The results showed that, under the chosen WTP ($37,663/QALYs), the probability of the suge arm being cost-effective compared with the placebo arm was 0% for both cCRT and sCRT population. However, when the WTP threshold increased to $50,000/QALYs, the probability of suge arm being economical was 0 and 50% for the cCRT and sCRT population, respectively. Given that the PAP was available, the probability of the suge arm being a cost-utility strategy was 100%.

### 3.4 Scenario analyses results

The results of each scenario analyses for cCRT and sCRT populations were displayed in Tables 4 and 5, respectively. Scenario analyses for cCRT population revealed that the model was most sensitive to assumptions regarding time horizon ($247,717), tumor histological type distribution ($89,347–90,675), discount in sugemalimab ($53,854–72,065), choice of subsequent immunotherapy ($80,728), mean duration of maintenance treatment ($55,153) and mean duration of subsequent immunotherapy ($91,277–91,775) (Table 4). However, no scenario could drag the ICUR below the $37,663 WTP threshold. Furthermore, the ICUR of suge arm remained surpass $37,663 except for scenarios that the discount in sugemalimab, choice of subsequent immunotherapy and the mean duration of maintenance treatment for the sCRT population (Table 5).

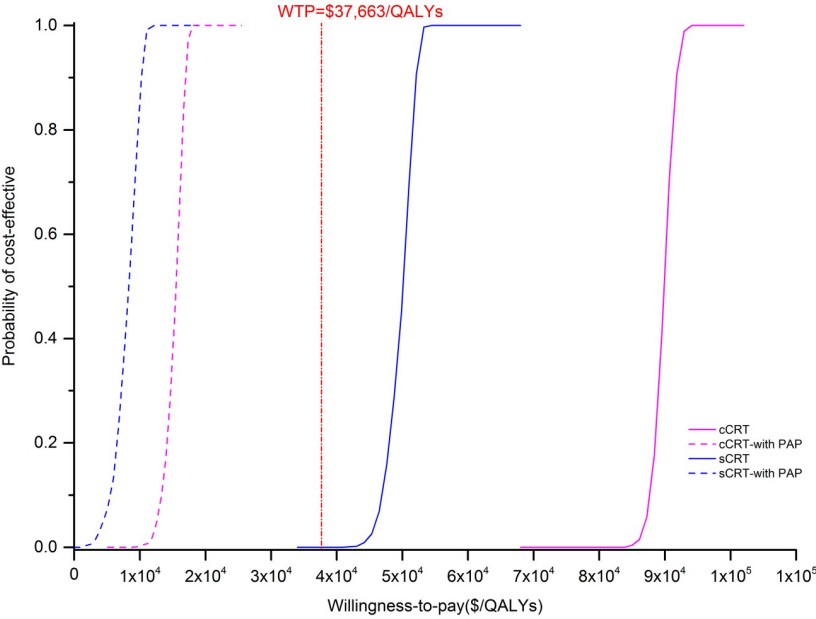

**Fig 7. Cost-effectiveness acceptability curves of sugemalimab vs. placebo with and without PAP.** PAP: Patient assistance program.

## 3.5 Model validation

Validation results for modeled PFS and OS data were presented in Table 6. The mPFS, mOS, and 1, 2, 3, 4, 5-year survival rate estimated by best-fitting distribution models matched the results that observed in GEMSTONE-301(cutoff date: March 1, 2022) [16,17], PACIFIC [29], PACIFIC-6 [35], Alliance A151812 [36] and START [37] clinical trials satisfactorily.

**Table 4. Results of scenario analyses for cCRT population.**

| Scenario | Cost suge | Cost placebo | Incremental cost | QALYs suge | QALYs placebo | Incremental QALYs | ICUR |
|---|---|---|---|---|---|---|---|
| **Time horizon** | | | | | | | |
| 5-year | 83,198 | 8,883 | 74,315 | 2.59 | 2.29 | 0.30 | 247,717 |
| **Tumor histological type** | | | | | | | |
| 100% SQ | 83,163 | 7,903 | 75,260 | 3.72 | 2.89 | 0.83 | 90,675 |
| 100% NSQ | 86,549 | 12,391 | 74,158 | 3.72 | 2.89 | 0.83 | 89,347 |
| **Discount in suge** | | | | | | | |
| 80% | 69,063 | 9,249 | 59,814 | 3.72 | 2.89 | 0.83 | 72,065 |
| 60% | 53,948 | 9,249 | 44,699 | 3.72 | 2.89 | 0.83 | 53,854 |
| **Choice of subsequent immunotherapy** | | | | | | | |
| Pembrolizumab | 88,802 | 21,798 | 67,004 | 3.72 | 2.89 | 0.83 | 80,728 |
| **Mean duration of maintenance treatment** | | | | | | | |
| 12 months | 55,026 | 9,249 | 45,777 | 3.72 | 2.89 | 0.83 | 55,153 |
| **Mean duration of subsequent immunotherapy** | | | | | | | |
| 6 months | 83,435 | 7,262 | 76,173 | 3.72 | 2.89 | 0.83 | 91,775 |
| 12 months | 83,669 | 7,909 | 75,760 | 3.72 | 2.89 | 0.83 | 91,277 |

cCRT, concurrent chemoradiotherapy; QALYs, quality-adjusted life years; ICUR, incremental cost utility ratio; suge, sugemalimab.

**Table 5. Results of scenario analyses for sCRT population.**

| Scenario | Cost suge | Cost placebo | Incremental cost | QALYs suge | QALYs placebo | Incremental QALYs | ICUR |
|---|---|---|---|---|---|---|---|
| **Time horizon** | | | | | | | |
| 5-year | 68,326 | 13,968 | 54,358 | 2.66 | 2.05 | 0.61 | 89,111 |
| **Tumor histological type** | | | | | | | |
| 100% SQ | 66,353 | 11,270 | 55,083 | 3.43 | 2.34 | 1.09 | 50,535 |
| 100% NSQ | 71,801 | 19,779 | 52,022 | 3.43 | 2.34 | 1.09 | 47,727 |
| **Discount in suge** | | | | | | | |
| 80% | 56,496 | 13,823 | 42,673 | 3.43 | 2.34 | 1.09 | 39,150 |
| 60% | 45,004 | 13,823 | 31,181 | 3.43 | 2.34 | 1.09 | 28,606 |
| **Choice of subsequent immunotherapy** | | | | | | | |
| Pembrolizumab | 75,428 | 37,499 | 37,929 | 3.43 | 2.34 | 1.09 | 34,797 |
| **Mean duration of maintenance treatment** | | | | | | | |
| 12 months | 49,137 | 13,823 | 35,314 | 3.43 | 2.34 | 1.09 | 32,398 |
| **Mean duration of subsequent immunotherapy** | | | | | | | |
| 6 months | 66,784 | 10,108 | 56,676 | 3.43 | 2.34 | 1.09 | 51,996 |
| 12 months | 67,145 | 11,482 | 55,663 | 3.43 | 2.34 | 1.09 | 51,067 |

sCRT, sequential chemoradiotherapy; QALYs, quality-adjusted life years; ICUR, incremental cost utility ratio; suge, sugemalimab.

## 4. Discussion

Till now, a number of trial-based economic assessments on durvalumab in post-chemoradiotherapy patients with unresectable stage III NSCLC from the Chinese healthcare system perspective have been reported [34,38,39]. The clinical data of these studies were all acquired from the 4-year survival updated PACIFIC clinical trial [40]. Sugemalimab, as the second domestic PD-L1 inhibitor approved by China for treating unresectable stage III NSCLC, represented a milestone for Chinese patients suffering from lung cancer. However, the high costs of this novel therapy might bring financial burden for the majority of middle- and low-income rural patients. It is an urgent agenda to evaluate the cost-effectiveness of sugemalimab consolidation therapy for lung cancer in the Chinse context. To the best of our knowledge, this is the first economic study on sugemalimab consolidation therapy for unresectable stage III NSCLC in China.

In this present study, we built a dynamic state transition model based on the latest data from Asian ethnicity population in GEMSTONE-301 study to estimate the cost-effectiveness of sugemalimab vs. placebo as consolidation therapy for unresectable stage III NSCLC in China. And we found that sugemalimab consolidation therapy was not cost-effective compared with placebo for sCRT and cCRT population at current WTP threshold ($37,663/QALYs). The probability of suge arm being cost-effective for cCRT and sCRT population increased with the raising of WTP, but the sCRT was more likely to be economic at a relatively lower threshold. However, when considering the price reduction (i.e. PAP), the suge arm exhibited favorable cost-effectiveness for both cCRT and sCRT population. The results of one-way deterministic and probabilistic sensitivity analysis both proved that the outcomes were generally robust. However, despite remaining above the WTP threshold, the ICUR is most influenced by the change in discount rate. This is because the discount rate reduces the benefit of the immune checkpoint inhibitor therapy, as the benefit may occur long time after immunotherapy.

Due to the immature of OS data in GEMSTONE-301, unseemly parametric model selection might overestimate or underestimate long-term survival outcomes beyond the study follow-up

**Table 6. Validation of modeled PFS and OS data using updated GEMSTONE-301 and external data.**

| Curve | Median, months (95% CI) | 1-year rate, % (95% CI) | 2-year rate, % (95% CI) | 3-year rate, % (95% CI) | 4-year rate, % (95% CI) | 5-year rate, % (95% CI) |
|---|---|---|---|---|---|---|
| **cCRT population** | | | | | | |
| **PFS** | | | | | | |
| **Suge arm** | | | | | | |
| Modeled | 15.4 (11.6–22.4) | 56.7 (49.0–63.2) | 40.7 (32.2–48.2) | 33.6 (23.0–42.7) | 30.0 (17.0–40.4) | 28.1 (12.8–39.5) |
| GEMSTONE-301 | 15.7 (9.0–24.4) | 55.9 (48.7–64.1) | 42.8 (34.9–52.5) | 34.9 (26.0–47.0) | NA | NA |
| PACIFIC | 16.9 (13.0–23.9) | 55.7 (51.0–60.2) | 45.0 (40.1–49.8) | 39.7 (34.7–44.7) | 35.0 (29.9–40.1) | 33.1 (28.0–38.2) |
| **Placebo arm** | | | | | | |
| Modeled | 10.2 (6.8–14.7) | 46.3 (35.4–56.2) | 29.8 (15.4–41.9) | 23.1 (5.4–38.0) | 20.0 (1.8–36.8) | 18.5 (0.4–36.4) |
| GEMSTONE-301 | 8.3 (5.8–24.8) | 41.7 (32.0–54.3) | 34.3 (24.3–48.5) | NA | NA | NA |
| PACIFIC | 5.6 (4.8–7.7) | 34.5 (28.3–40.8) | 25.1 (19.3–31.2) | 20.8 (15.3–26.9) | 19.9 (14.4–26.1) | 19.0 (13.6–25.2) |
| **OS** | | | | | | |
| **Suge arm** | | | | | | |
| Modeled | 38.5 (31.5, NR) | 85.4 (81.3–90.9) | 66.0 (60.9–81.0) | 52.4 (45.6–78.7) | 44.4 (35.9–78.1) | 40.1 (29.2–78.0) |
| GEMSTONE-301 | NR (28.2, NR) | 85.4 (80.5–91.1) | 66.3 (58.4–74.9) | 54.1 (43.5–66.9) | NA | NA |
| PACIFIC | 47.5 (38.1–52.9) | 83.1 (79.4–86.2) | 66.3 (61.8–70.4) | 56.7 (52.0–61.1) | 49.7 (45.0–54.2) | 42.9 (38.2–47.4) |
| **Placebo arm** | | | | | | |
| Modeled | 30.8 (22.4–42.0) | 80.9 (73.1–87.2) | 59.4 (47.4–68.4) | 44.7 (29.7–55.1) | 34.6 (19.2–46.1) | 27.4 (12.9–39.8) |
| GEMSTONE-301 | 32.4 (20.6, NR) | 84.8 (78.7–93.6) | 57.6 (46.3–71.6) | 19.8 (4.3–90.1) | NA | NA |
| PACIFIC | 29.1 (22.1–35.1) | 74.6 (68.5–79.7) | 55.3 (48.6–61.4) | 43.6 (37.1–49.9) | 36.3 (30.1–42.6) | 33.4 (27.3–39.6) |
| **sCRT population** | | | | | | |
| **PFS** | | | | | | |
| **Suge arm** | | | | | | |
| Modeled | 7.7 (5.6–10.9) | 39.2 (29.0–47.6) | 24.7 (15.8–32.7) | 19.7 (10.4–29.0) | 17.6 (7.3–27.8) | 16.7 (5.8–27.3) |
| GEMSTONE-301 | 8.1 (3.4–10.6) | 38.8 (30.1–52.0) | 30.5 (21.3–42.2) | 18.1 (10.8–31.6) | NA | NA |
| PACIFIC-6 | 10.9 (7.3–15.6) | 49.6 (39.5–58.9) | NA | NA | NA | NA |
| **Placebo arm** | | | | | | |
| Modeled | 3.4 (2.7–5.6) | 15.4 (8.4–46.0) | 9.6 (4.3–44.8) | 7.8 (3.2–44.4) | 7.0 (2.7–44.2) | 6.5 (2.4–44.1) |
| GEMSTONE-301 | 4.1 (2.1–6.1) | 12.7 (4.8–28.4) | 7.6 (3.0–25.1) | NA | NA | NA |
| Alliance A151812 | 8.0 (6.8–10.5) | 35.9 (27.6–46.6) | 16.6 (10.6–26.1) | 11.7 (6.6–20.7) | 8.9 (4.5–17.7) | 6.7 (2.8–16.3) |
| **OS** | | | | | | |
| **Suge arm** | | | | | | |
| Modeled | 44.8 (32.9-NR) | 85.7 (80.4–91.9) | 71.3 (63.2–79.8) | 58.7 (46.7–70.0) | 48.1 (33.1–61.9) | 39.3 (23.5–55.3) |
| GEMSTONE-301 | NR (31.9, NR) | 89.1 (82.7–96.1) | 70.7 (63.8–83.2) | 59.0 (48.8–72.6) | NA | NA |
| PACIFIC-6 | 25.0 (25.0-NR) | 84.1 (75.6–89.9) | 69.8 (55.8–80.2) | NA | NA | NA |
| **Placebo arm** | | | | | | |
| Modeled | 26.3 (19.6–36.0) | 83.0 (72.7–92.1) | 55.0 (38.9–68.6) | 35.2 (19.0–50.0) | 23.4 (9.6–38.3) | 16.4 (5.4–30.6) |
| GEMSTONE-301 | 24.1 (19.5, NR) | 80.2 (68.6–93.4) | 53.7 (45.2–76.7) | 43.7 (28.4–63.3) | NA | NA |

*(Continued)*

**Table 6.** (Continued)

| Curve | Median, months (95% CI) | 1-year rate, % (95% CI) | 2-year rate, % (95% CI) | 3-year rate, % (95% CI) | 4-year rate, % (95% CI) | 5-year rate, % (95% CI) |
|---|---|---|---|---|---|---|
| START | 24.6 (18.8–33.0) | 73.7 (66.7–81.6) | 50.7 (42.5–60.4) | 37.0 (29.1–47.2) | 22.7 (14.3–35.8) | about 9.8 |

cCRT, concurrent chemoradiotherapy; sCRT, sequential chemoradiotherapy; mPFS, median progression-free survival; mOS, median overall survival; NR: Not reached; NA, not applicable; Suge, sugemalimab.

period. To minimize the potential bias introduced by model selection, we introduced both internal and external validations. The selection of best-fitting parametric distributions for PFS and OS curves in each arm was the result of a comprehensive analysis, taking into account not only the AIC, BIC and median survival time but also the long-term survival rate, as well as clinical rationality. The Gompertz distribution model for PFS curve in placebo arm estimated a mPFS of 10.2 months for cCRT population, which seemed to be a significant difference comparing with that of the 8.3 month in GEMSTONE-301. Judging only by the mPFS, the Gompertz distribution model did not seem to be the best fit choice. However, the 5-year survival rate in placebo arm of cCRT population estimated by Gompertz model (18.5 vs. 19.0%) was closest to that of PACIFIC clinical trial (i.e. external data) among all distribution models. The Log-normal and mixture cure (Gamma) model estimated a 5-year survival rate of 40.1 vs. 27.4% for suge arm and placebo arm in cCRT respectively, which was approximate to that of PACIFIC trial (42.9 vs. 33.4%). By comparing the baseline characteristics of GEMSTONE-301 and PACIFIC, it was not difficult to conclude that the GEMSTONE-301 study population was more refractory, with higher proportion of patients with stage IIIB/IIIC, squamous cell carcinoma, and ECOG PS = 1, resulting a relatively lower survival rate. For sCRT population, the Log-logistic predicted a 5-year overall survival rate of 16.4% for placebo arm and its extrapolation part matched the K-M curve in START trial [37] was selected as a best fit distribution. In addition, Log-logistic projected an 8-year survival rate of 6.0%, which was consistent with the results of the Alliance A151812 study [36]. As for the suge arm in sCRT, the mixture cure model (Gamma) estimated a 5-year survival rate was 39.3%, lower than that of 40.1% in cCRT, which was in line with the conclusion of a previously reported Meta-analysis [41]. As NICE DSU TSD 14 recommended [42] that the use of the same type of parametric model for individual intervention groups to evite drastically different shapes of survival curves, unless there is a reasonable explanation. Although we selected different types of parametric models for individual treatment arms, deviating from this guidance, this selection exhibited superior external and internal validity and plausibility.

There were several limitations in this study that should to be noted. Firstly, the state-transition model used in this present study was that PFS, OS, and OS data were immature, which might affect the simulation of the long-term survival curves and increase model uncertainty. Secondly, the lack of HRQoL data from China, which has different patient preferences and healthcare settings from many developed countries, might bring bias in economic evaluations. It should be noted that the HRQoL might differ in post-cCRT versus post-sCRT patients due to the side effects and adherence. However, the sensitivity analysis indicated that the utility of both PFS and PD fluctuated in given ranges did not affect the robust of model outcomes. Thirdly, while the base-case analysis represented a likely set of therapeutic regimens in China, in clinical practice, different patients might have different preferences for subsequent treatment options. Despite the limitations of this present study, the uncertainty analysis demonstrated the robustness of our model outcomes. Therefore, it might provide valuable reference information for clinical treatment decisions and policy-makers.

## 5. Conclusions

From the perspective of Chinese healthcare system, sugemalimab consolidation therapy vs. placebo after cCRT or sCRT was not cost-effective for patients with unresectable stage III NSCLC. However, the use of the Patient Assistance Program (PAP) should be considered as a favorable cost-effective treatment option in China.

## Supporting information

**S1 Fig. Tornado diagram of one-way deterministic sensitivity analysis (DSA) of sugemalimab vs. placebo. DSA for cCRT population with PAP.** cCRT, concurrent chemoradiotherapy; PAP: Patient assistance program.
(TIF)

**S2 Fig. Tornado diagram of one-way deterministic sensitivity analysis (DSA) of sugemalimab vs. placebo. DSA for sCRT population with PAP.** sCRT, sequential chemoradiotherapy; PAP: Patient assistance program.
(TIF)

**S1 Table. AIC and BIC of parametric distributions for cCRT and sCRT curves.** *, best fitted model; cCRT, concurrent chemoradiotherapy; sCRT, sequential chemoradiotherapy; Suge: Sugemalimab; PC; AIC, Akaike information criterion; BIC, Bayesian information criterion; PFS, progression-free survival; OS, overall survival; Suge, sugemalimab.
(DOCX)

**S1 Dataset. Raw data.**
(RAR)

## Author Contributions

**Investigation:** Wei Li.

**Methodology:** Wei Li.

**Writing – original draft:** Wei Li.

**Writing – review & editing:** Wei Li.

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
