## [Decision Letter · Decision Letter 0]

16 Mar 2023

PONE-D-23-01774A trial-based cost-effectiveness analysis of sugemalimab vs. placebo as consolidation therapy for unresectable stage III NSCLC in ChinaPLOS ONE

Dear Dr. Li,

Thank you for submitting your manuscript to PLOS ONE. After careful consideration, we feel that it has merit but does not fully meet PLOS ONE’s publication criteria as it currently stands. Therefore, we invite you to submit a revised version of the manuscript that addresses the points raised during the review process.

On this occasion, I apologize that the review process took a long time. Finding the right reviewers for this type of manuscript is very challenging.

While I encourage you to respond to all comments, it is especially important to respond to ethical comment.

We look forward to receiving your revised manuscript.

Kind regards,

Mirosława Püsküllüoğlu, MD, PhD

Academic Editor

PLOS ONE

Journal Requirements:

Reviewers' comments:

Reviewer's Responses to Questions

**Comments to the Author**

1. Is the manuscript technically sound, and do the data support the conclusions?

Reviewer #1: Partly

Reviewer #2: Yes

2. Has the statistical analysis been performed appropriately and rigorously? 

Reviewer #1: Yes

Reviewer #2: Yes

3. Have the authors made all data underlying the findings in their manuscript fully available?

Reviewer #1: No

Reviewer #2: Yes

4. Is the manuscript presented in an intelligible fashion and written in standard English?

Reviewer #1: Yes

Reviewer #2: Yes

5. Review Comments to the Author

Reviewer #1: The analysis described is important to inform country and regional level decisions on affordability of treatment regimens for NSCLC, drug access programming, and negotiations with treatment providers.

Basing off interim analysis from only 5 years of followup in an ongoing trial (NCT03728556), however, limits the reliability of the study's extrapolation of costs and outcomes over a horizon of 21 years. Though the authors describe fitness testing for various decay models, it might be more appropriate to provide the results based on the shorter horizon of 10 years in the main finding.

The methodological approach is mostly standard, and the analytical approach is adequately elaborated. A couple of clarifications are needed on some definitions and descriptions as highlighted in the attached file. Of note, it would be good to state that the outcomes were simulated (and not just derived) based on the trial's interim survival curves. Although the framework for estimation of direct costs of medication (drugs and dosages) post CRT are described, there is little described in terms of other inpatient and outpatient care costs. Were there any estimates applied of future service utilisation and where were these derived from? Were there estimates of inpatient care? How were palliative care costs estimated? Where progression occurs, two different treatment approaches are presented but no associated paths and probabilities are shown in the decision tree making it difficult to figure out how these associated costs were computed.

On the results, the predicted survival curves (Fig 2 and 3) and the rates provided in table 6 from the model don't have associated predictive bounds, and the authors may need to explain this choice of visualisation.

On the discussion section it may be worthwhile noting if HRQoL for Stage III patients might differ post cCRT versus post sCRT, and whether this would inform interpretation of results, considering side effects and adherence, which were not included in this model. It may be useful to discuss the implication of the results of the one way sensitivity on the discount rate (here I'm refering to the tornado diagrams). Is this large influence a result of the long horizon selected? What does this imply for the decisionmaker?

With regards to the data availability question (Question 3 on reviewer panel), for a more transparent / reproducible markov model, the authors may consider tabulating the actual state transition probabilities that were extracted from the interim results of the trial and that have been used. Further, presenting a table of the hazard ratios at multiple time points allows one to judge the appropriateness of the selected parametric models.

With these comments and others addressed, I believe the manuscript is in a good place to inform the practice of management of NSCLC.

Thanks

Reviewer #2: Dear authors, here are my comments and concern on the manuscript.

1. This manuscript titled “A trial-based cost-effectiveness analysis of sugemalimab vs. placebo as Consolidation therapy for unresectable stage III NSCLC in China” is a very vital and will add relevant findings in the context of China and beyond.

2. However, the ethical issue is not well addressed, simply stated that N/A. The reviewer does not feel that the manuscript meets the journal (PLOS ONE)’s ethical requirements. PLOS ONE requires that research meets all applicable standards for the ethics of experimentation and research integrity. For example, if standard drug is available there, using placebo may not be an ethical.

3. In the abstract, some components were missed. For example, the background is not stated in this section. In this section, the methods part is not comprehensive and exhaustive. For example, what types of costs (direct medical costs and non-medical costs, indirect cost, and intangible costs) the authors considered for this economic evaluation.

4. The authors stated that they estimated costs and health utility in the methods part. Does it cost effective analysis or cost utility analysis?

5. Moreover, some sentences are too long and not clear, so that they need reconstruction. Hence, the manuscript needs minor revision and clarification on its ethical aspect before publication.

6. PLOS authors have the option to publish the peer review history of their article (what does this mean?). If published, this will include your full peer review and any attached files.

Reviewer #1: **Yes: **Peter Nguhiu

Reviewer #2: **Yes: **Gebeyehu Tsega Nebeb

---

## [Author Response · Author response to Decision Letter 0]

17 Apr 2023

Dear Mirosława Püsküllüoğlu and reviewers,

Thank you for your letter and the reviewers’ comments on our manuscript entitled “A TRIAL-BASED COST-EFFECTIVENESS ANALYSIS OF SUGEMALIMAB VS. PLACEBO AS CONSOLIDATION THERAPY FOR UNRESECTABLE STAGE III NSCLC IN CHINA”(manuscript number: PONE-D-23-01774). Those comments are very helpful for revising and improving our paper, as well as the important guiding significance to other research. We have studied the comments carefully and made corrections which we hope meet with approval. The main corrections are in the manuscript and the responds to the reviewers’ comments are as follows (the responses are highlighted in blue).

Response to the reviewers’ comments:

Reviewer #1:

The analysis described is important to inform country and regional level decisions on affordability of treatment regimens for NSCLC, drug access programming, and negotiations with treatment providers.

1. Basing off interim analysis from only 5 years of follow up in an ongoing trial (NCT03728556), however, limits the reliability of the study's extrapolation of costs and outcomes over a horizon of 21 years. Though the authors describe fitness testing for various decay models, it might be more appropriate to provide the results based on the shorter horizon of 10 years in the main finding.

Response: Thanks for this very useful comment. In our base-case analysis, we set the time horizon to 10 years in our revised version.

2. The methodological approach is mostly standard, and the analytical approach is adequately elaborated. A couple of clarifications are needed on some definitions and descriptions as highlighted in the attached file. 

2.1 Of note, it would be good to state that the outcomes were simulated (and not just derived) based on the trial's interim survival curves. 

Response: Thank you for your suggestion. We have corrected our statement that the outcomes are simulated based on trial's interim survival curves.

2.2 Although the framework for estimation of direct costs of medication (drugs and dosages) post CRT are described, there is little described in terms of other inpatient and outpatient care costs. 

Response: Thank you for your comment. We define in detail the costs of routine follow-up (i.e. cost of outpatient physician visit, hospitalization, and laboratory tests), best supportive care (BSC) and terminal care in Table 2. The BSC referred to the intervention of clinical symptoms caused by cancer, including anti-inflammatory treatment, analgesic treatment, antiemetic treatment, thoracic and abdominal puncture decompression, blood transfusion and nutritional support. The terminal care cost referred to the cost of palliative end-of-life.

2.3 Were there any estimates applied of future service utilisation and where were these derived from? 

Response: Thank you for your comment. The consolidation therapy of sugemalimab in post-CRT population can prolong the PFS and OS, thus may reduce utilisation of health services in the future. We do not consider this in our study, due to the lack of associated data. In addition, we may not understand what you mean. If there is any discrepancy in my response, please inform me, and I will modify it in the next revision.

2.4 Were there estimates of inpatient care? 

Response: Thank you for your suggestion. The costs of inpatient care such as the costs of routine follow-up, best supportive care (BSC) and terminal care are included in our model analysis.

2.5 How were palliative care costs estimated? Where progression occurs, two different treatment approaches are presented but no associated paths and probabilities are shown in the decision tree making it difficult to figure out how these associated costs were computed.

Response: Thank you for your suggestion. The proportion of subsequent immunotherapy and detailed treatment strategies varied with tumor histology type (squamous or non- squamous) at progression was shown in Fig 4 in revised version.

3. On the results, the predicted survival curves (Fig 2 and 3) and the rates provided in table 6 from the model don't have associated predictive bounds, and the authors may need to explain this choice of visualisation.

Response: Thank you for your suggestion. We have given 95% confidence interval (95% CI) of estimated values in our revised version in Table 6. In the revised Figures (Figs 2 and 3), we add external validation curve for facilitate justify the rationality of curve extrapolation by visual inspection. The fitted parametric survival model curves were overlaid on the Kaplan-Meier curves to assess how closely the modeled curves matched the observed nonparametric survival estimates. Overall, all fitted curves perform a good fit to the survival curves (PFS and OS) within the follow-up time, but the curves diverge beyond follow-up period. Therefore, we focus on visual inspection to pick out modeled curves that better fit external validation curve beyond the follow up time. The reasons for choosing the best fitting distributions are detailed in the Discussion section. 

4. On the discussion section it may be worthwhile noting if HRQoL for Stage III patients might differ post cCRT versus post sCRT, and whether this would inform interpretation of results, considering side effects and adherence, which were not included in this model. It may be useful to discuss the implication of the results of the one way sensitivity on the discount rate (here I'm refering to the tornado diagrams). Is this large influence a result of the long horizon selected? What does this imply for the decisionmaker?

Response: Thank you for your suggestion. We have discussed the concern regarding HRQoL in our discussion section in our revised version. And we also explain why the ICUR is most sensitive to the discount rate in discussion section. This is because the discount rate reduces the benefit of the immunotherapy, as the benefit may occur long time after immunotherapy. When setting different time horizon (i.e. 5-year, 10-year) in this analytic model, the discount rate is still the most influential parameter on ICUR, but this do not change the economic outcomes. So, this large influence is not a result of the long horizon selected. For decision makers, a lower discount rate instead of the usually applied 5% in China when the benefit of intervention is sustained over a long period may be more appropriate. 

5. With regards to the data availability question (Question 3 on reviewer panel), for a more transparent / reproducible markov model, the authors may consider tabulating the actual state transition probabilities that were extracted from the interim results of the trial and that have been used. Further, presenting a table of the hazard ratios at multiple time points allows one to judge the appropriateness of the selected parametric models.

Response: Thank you for your suggestion. We have presented the state transition probabilities and the hazard ratios at multiple time points in raw data.

Reviewer #2:

1. This manuscript titled “A trial-based cost-effectiveness analysis of sugemalimab vs. placebo as Consolidation therapy for unresectable stage III NSCLC in China” is a very vital and will add relevant findings in the context of China and beyond.

Response: Thank you for your positive comment.

2. However, the ethical issue is not well addressed, simply stated that N/A. The reviewer does not feel that the manuscript meets the journal (PLOS ONE)’s ethical requirements. PLOS ONE requires that research meets all applicable standards for the ethics of experimentation and research integrity. For example, if standard drug is available there, using placebo may not be an ethical.

Response: Thanks for this very useful comment. This study is based on previously conducted studies and does not contain any studies with human participants or animals performed by any of the authors. We have added this statement in our Ethics Statement.

3. In the abstract, some components were missed. For example, the background is not stated in this section. In this section, the methods part is not comprehensive and exhaustive. For example, what types of costs (direct medical costs and non-medical costs, indirect cost, and intangible costs) the authors considered for this economic evaluation.

Response: Thank you for your comment. We have updated our abstract in our revised version. 

4. The authors stated that they estimated costs and health utility in the methods part. Does it cost effective analysis or cost utility analysis?

Response: Thank you for your comment. This is a cost-utility analysis (CUA) and we have corrected it in our revised version.

5. Moreover, some sentences are too long and not clear, so that they need reconstruction. Hence, the manuscript needs minor revision and clarification on its ethical aspect before publication.

Response: We have reconstructed the long and unclear sentences in our revised version and highlighted in yellow. And we also clarified our ethical statement.

---

## [Decision Letter · Decision Letter 1]

19 May 2023

A trial-based cost-utility analysis of sugemalimab vs. placebo as consolidation therapy for unresectable stage III NSCLC in China

PONE-D-23-01774R1

Dear Dr. Li,

We’re pleased to inform you that your manuscript has been judged scientifically suitable for publication and will be formally accepted for publication once it meets all outstanding technical requirements.

Kind regards,

Mirosława Püsküllüoğlu, MD, PhD

Academic Editor

PLOS ONE

Additional Editor Comments (optional):

Reviewers' comments:

Reviewer's Responses to Questions

**Comments to the Author**

1. If the authors have adequately addressed your comments raised in a previous round of review and you feel that this manuscript is now acceptable for publication, you may indicate that here to bypass the “Comments to the Author” section, enter your conflict of interest statement in the “Confidential to Editor” section, and submit your "Accept" recommendation.

Reviewer #1: All comments have been addressed

Reviewer #3: All comments have been addressed

2. Is the manuscript technically sound, and do the data support the conclusions?

Reviewer #1: Yes

Reviewer #3: Yes

3. Has the statistical analysis been performed appropriately and rigorously? 

Reviewer #1: Yes

Reviewer #3: Yes

4. Have the authors made all data underlying the findings in their manuscript fully available?

Reviewer #1: Yes

Reviewer #3: Yes

5. Is the manuscript presented in an intelligible fashion and written in standard English?

Reviewer #1: Yes

Reviewer #3: Yes

6. Review Comments to the Author

Reviewer #1: Thank you for the well detailed responses, that made it easy to follow the changes made. The explanations and deductions made are consistent with the results presented, and the revised manuscript provides better context for interpretation, to the data.

One very interesting revision, which I'm not very confident about, is the decision to use the phrase 'incremental cost utility ratio' instead of the more familiar term 'incremental cost effect ratio'. While I agree with my co-reviewer who correctly noted that this analysis would be classified as a cost utility analysis, I am not familiar with any previous use of the specific phrase 'incremental cost utility ratio' and would have maintained the more common term and its abbreviation (ICER) for the benefit of ease of comprehension by the decision makers that are the targeted audience.

Reviewer #3: Thank you for submitting a sound revision of your manuscript.

All comments have bee thoroughly addressed.

7. PLOS authors have the option to publish the peer review history of their article (what does this mean?). If published, this will include your full peer review and any attached files.

Reviewer #1: **Yes: **Peter Nguhiu

Reviewer #3: No

---

## [Editor Report · Acceptance letter]

23 May 2023

PONE-D-23-01774R1 

A trial-based cost-utility analysis of sugemalimab vs. placebo as consolidation therapy for unresectable stage III NSCLC in China 

Dear Dr. Li:

I'm pleased to inform you that your manuscript has been deemed suitable for publication in PLOS ONE. Congratulations! Your manuscript is now with our production department. 

Kind regards, 

on behalf of

Dr. Mirosława Püsküllüoğlu 

Academic Editor

PLOS ONE